# Structural Characteristics of High-Mobility Group Proteins HMGB1 and HMGB2 and Their Interaction with DNA

**DOI:** 10.3390/ijms24043577

**Published:** 2023-02-10

**Authors:** Tatiana Y. Starkova, Alexander M. Polyanichko, Tatiana O. Artamonova, Anna S. Tsimokha, Alexey N. Tomilin, Elena V. Chikhirzhina

**Affiliations:** Laboratory of Molecular Biology of Stem Cells, Institute of Cytology of the Russian Academy of Sciences, Tikhoretsky Av. 4, 194064 St. Petersburg, Russia

**Keywords:** non-histone chromosomal proteins HMGB1 and HMGB2, DNA–protein interactions, circular dichroism, mass spectrometry, post-translational modifications (PTMs)

## Abstract

Non-histone nuclear proteins HMGB1 and HMGB2 (High Mobility Group) are involved in many biological processes, such as replication, transcription, and repair. The HMGB1 and HMGB2 proteins consist of a short N-terminal region, two DNA-binding domains, A and B, and a C-terminal sequence of glutamic and aspartic acids. In this work, the structural organization of calf thymus HMGB1 and HMGB2 proteins and their complexes with DNA were studied using UV circular dichroism (CD) spectroscopy. Post-translational modifications (PTM) of HMGB1 and HMGB2 proteins were determined with MALDI mass spectrometry. We have shown that despite the similar primary structures of the HMGB1 and HMGB2 proteins, their post-translational modifications (PTMs) demonstrate quite different patterns. The HMGB1 PTMs are located predominantly in the DNA-binding A-domain and linker region connecting the A and B domains. On the contrary, HMGB2 PTMs are found mostly in the B-domain and within the linker region. It was also shown that, despite the high degree of homology between HMGB1 and HMGB2, the secondary structure of these proteins is also slightly different. We believe that the revealed structural properties might determine the difference in the functioning of the HMGB1 and HMGB2 as well as their protein partners.

## 1. Introduction

Proteins HMGB1 and HMGB2 belong to a large family of HMG proteins characterized by high electrophoretic mobility (High Mobility Group). HMG proteins are the largest group of nuclear proteins after histones. According to their structure and functions, these proteins are divided into three families: HMGA (formerly known as HMGI/Y), HMGN (formerly known as HMG14/17), and HMGB domain proteins (HMGB1-4, formerly known as HMG1/2) [1,2,3,4]. HMGA proteins interact with AT-rich DNA regions and are involved in the regulation of chromatin structure and gene transcription [5,6]. HMGNs bind to nucleosomes and take part in transcription initiation. However, these proteins are not part of the transcription complex [7,8]. HMGB domain proteins interact with DNA through DNA binding domains. It is worth noting that many transcription factors have domains homologous to HMGB1 DNA-binding domains. Proteins of the HMGB family are the most common and studied proteins among the HMG proteins. All proteins of this family are characterized by the presence of a structural and functional motif known as the HMGB domain (~80 aa) [4,9,10,11,12]. Proteins HMGB1 and HMGB2 have two such domains and, like other members of this family, demonstrate non-sequence-specific DNA binding [2,4,9]. A characteristic feature of these proteins is their ability to recognize and bind structurally disordered DNA regions [4,13]. It is known that both proteins are actively involved in the regulation of basic nuclear processes, such as transcription, replication, recombination, and DNA repair [2,4,9]. In addition, HMGB domains homologous to the domain of the HMGB1 protein were identified as DNA-binding elements in many transcription factors [2,3,9].

The HMGB1 and HMGB2 proteins are very similar in structure and amino acid sequence. Both consist of a short N-terminal region, two DNA-binding domains, A and B, and a C-terminal sequence of glutamic and aspartic acids. These proteins are evolutionarily conserved (Figure 1). For example, in the bull, rat, mouse, pig, and human, they are 95–99% identical [14]. The most notable difference between HMGB1 and HMGB2 resides in the length of the C-terminal region, which consists of 30 and 20 a.a. residues, respectively, and do not have a particular structural organization. These negatively charged C-terminal sequences modulate the interaction of proteins with nucleic acids and other proteins [4,12,15,16,17,18,19,20,21,22,23,24,25]. At the same time, having highly homologous amino acid sequences, the HMGB1 and HMGB2 proteins should have other features that determine their functional differences. Such features can comprise, for example, the different pattern of their PTMs and the structural organization of their polypeptide chains.

The variety of functions performed by proteins is associated with their localization in the cellular/intercellular space [4,23,26,27,28]. Depending on its PTMs and redox status, HMGB1 can leave the nucleus, move into the cytoplasm, and exit to the intercellular space, where it stimulates the immune response and functions as a signaling molecule in response to damage to cell integrity and necrosis. HMGB1 is subject to such modifications as acetylation, phosphorylation, methylation, glycosylation, and poly-ADP-ribosylation [3]. Phosphorylation and acetylation of the HMGB1 protein are primarily required for its interaction with DNA. Three cysteine residues located in the DNA-binding domains of the protein are oxidized [29]. The protein changes its localization in the cell space depending on the redox state of cysteine residues and the presence and location of acetylated, ADP-ribosylated [30], and methylated [31] sites. Oxidation of cysteines C23 and C45, followed by the formation of disulfide bridges, drives the protein first into the cytosol and then into the extracellular space (the C residues are oxidized under the action of reactive oxygen species) [29,32]. Depending on the redox state, extracellular HMGB1 can (1) act as a signaling molecule by activating the MAPKs (infarct, inflammation, and post-infarct remodeling [27]), NF-kB (thrombosis [33], urothelial carcinoma [34], inflammation during tissue damage [35]) and phosphoinositide-3-kinase/AKT signaling pathways (inflammation process of lungs or liver due to epilepsy, heart disease, cancer, diabetes [27,36,37]), (2) take part in the regulation of cell migration (regeneration, tissue damages [27,38,39]), or (3) participate in the immune response and synthesis of anti-inflammatory cytokines and chemokines (fibrosis [27], type 1 diabetes, myocardium, cancers [29,40,41,42,43,44]). The close composition of the amino acid sequences of the HMGB1 and HMGB2 proteins suggests that similar mechanisms underlie the functioning of HMGB2. To date, a large amount of data has been accumulated, and many works have been devoted to the role of the nuclear and extranuclear HMGB1 protein, while the functions of HMGB2 in the cytoplasm, on the membrane, and outside the cell are practically not studied. The PTMs of the HMGB2 protein presented in the literature are described mainly by analogy with the PTMs of HMGB1.

Having highly homologous amino acid sequences, the HMGB1 and HMGB2 proteins should have other features that determine their functional differences. Such features can include, for example, the nature/pattern of their PTMs and the structural organization of their polypeptide chains. However, currently, there is a lack of information about the differences between their PTMs and secondary structures. Here, we report new potential sites of PTMs, which have been revealed in HMGB1 and HMGB2. In this work, we also performed a comparative analysis of PTM patterns and secondary structures of the HMGB1 and HMGB2 proteins.

## 2. Results and Discussion

For the identification of structural characteristics of proteins and their complexes with DNA and also PTMs, nuclear HMGB1 and HMGB2 were prepared according to the approach described in Materials and Methods. Proteins were isolated from calf thymus by extraction with 5% perchloric acid followed by precipitation with 6 volumes of acidified acetone at −20 °C. Separation of the protein fraction into individual components was performed using ion exchange chromatography on an FPLC system. Results are presented in Appendix A).

### 2.1. Secondary Structure of HMGB1 and HMGB2 Proteins

The secondary structure of a protein is one of the basic characteristics that determine its biological functioning and physicochemical properties. HMG proteins are often compared with natively unfolded proteins due to their high structural flexibility [45,46,47,48]. To analyze possible differences in the structural properties of the HMGB1 and HMGB2, we obtained their CD spectra in the far ultraviolet region under various conditions that affect their secondary structure (Figure 2). In a neutral solvent (water, pH 6.0), both proteins show similar spectra typical for the α-helical and random coil conformations. Quantitative analysis of the spectra shows that under these conditions, HMGB1 contains more α-helical regions (~25%) compared to HMGB2 (15%). One of the characteristic indicators of the structural flexibility of a protein is its ability to change the secondary structure in the presence of alcohol (typically ethanol or trifluoroethanol) [49,50]. To estimate the ability of HMGB1 and HMGB2 to form additional α-helical regions, we analyzed their CD spectra in the presence of alcohol in several concentrations (Figure 2). The changes in solvent induced a significant increase in the intensity of negative CD bands for both proteins. The resulting α-helicity was estimated as 30% and 40% for HMGB1 and HMGB2, respectively.

Increasing the ionic strength of the solution to 1.5 M NaCl leads to the electrostatic screening of charged groups, resulting in suppression of their Coulombic attraction/repulsion, rearrangements of bound counterions, and inducing structural transitions in proteins [51]. Indeed, the CD spectra of HMGB1 and HMGB2 have deeper minima in the presence of higher NaCl concentrations (Figure 2). Surprisingly, these two proteins demonstrate different CD spectra, reflecting different conformations at high salt concentrations. The CD spectrum of HMGB2 is characterized by deeper negative bands, indicating a higher content of α-helical regions: ~70% in the HMGB2 vs. ~40% in HMGB1.

As shown earlier [17,52], the length of the acidic tail in HMGB1 is sufficient to interact with the adjacent DNA-binding B-domain of the same protein molecule. This interaction might have two consequences: (1) the DNA-binding B-domain of the HMGB1 blocked by the negatively charged tail cannot interact with DNA, and (2) the tail itself cannot establish intermolecular contacts. In the case of the HMGB2, the shorter tail is not long enough for effective intramolecular interaction, which allows both HMGB domains to bind DNA and stimulates interaction with other proteins via the C-terminal fragment. Most likely, in HMGB1, the binding of the C-terminal sequence to the HMGB domain, to some extent, limits the conformational flexibility of the latter. In HMGB2, on the contrary, both HMGB domains retain their conformational flexibility, which leads to an increased degree of α-helicity of the protein both in 1.5 M NaCl and 80% ethanol solutions. The changes in α-helicity of HMGB1 and HMGB2 proteins depending on the solvent shown by us are presented in Table 1. We also performed secondary structure analysis using the K2D3 algorithm [53,54]. The obtained results are in very good agreement with those obtained with the above equation, with deviations less than 5%.

The above differences in the secondary structure of the HMGB1 and HMGB2 chromosomal proteins can be due to at least two reasons: different lengths of the disordered negatively charged C-terminal regions or different nature of PTMs in these proteins, which can significantly modify the physicochemical properties of the polypeptide chain. To test both hypotheses, we conducted a comparative analysis of the PTMs of the HMGB1 and HMGB2 proteins and studied their interaction with DNA, which is the main binding target in the cell nucleus.

### 2.2. Post-Translational Modifications of HMGB1 and HMGB2 Proteins in the Calf Thymus

The HMGB1 and HMGB2 PTMs were analyzed using MALDI mass spectrometry. Appendix A shows the obtained mass spectra of the studied samples. The characteristics of the peptides identified in the samples are presented in Table 2 and Appendix A. Identified potential sites of PTMs are shown in Figure 3. According to the published data, HMGB1 is subject to such modifications as acetylation (Ac), phosphorylation (P), methylation (Met), glycosylation, and poly-ADP-ribosylation [4]. Although there is not enough experimental data on HMGB2 modifications, one could expect that the HMGB1 and HMGB2 have similar PTMs due to the similarity of their primary structures. However, our results suggest that despite the high identity of the amino acid sequences of the proteins, the nature and location of the PTMs identified in the HMGB1 and HMGB2 are surprisingly different (Table 2 and Appendix A). In HMGB1, PTMs are located mainly within DNA-binding domains (predominantly in the A domain) and the linker region, whereas in HMGB2, PTMs are concentrated within the B domain and linker.

Since there are very little data on HMGB2 modifications, we set ourselves a more general task—to reveal the modifications possible for these proteins and the regions of their location in the HMGB1 and HMGB2 proteins and compare them with each other. If there are several identified potential modification positions in a certain site, the MALDI-FT-ICR-MS analysis we used does not allow us to accurately identify the site where this modification occurred. The experimental results presented in this table reflect the areas of the amino acid sequence in which these modifications are present. Since we can only indicate the area, we did not highlight specific modification sites in the table. Based on the comparison of literature data with our experimental data, we constructed an approximate map of the position of potential modifications (Figure 3 and Appendix A). Further studies are certainly required to accurately determine the position of all identified modifications. We have tried to be as careful as possible in the description of particular modification sites, indicating them as potential or possible sites. Appendix A presents the data obtained after analysis using the Protein Prospector program. Figure 3 and Appendix A are maps of suggested potential modifications based on our comparative analysis of the complete data set. Although the obtained results can be further improved, they allow us to conclude that the regions where the revealed modifications are located are different in the HMGB1 and HMGB2 proteins, and this may indicate the different nature of the binding of these proteins to DNA and other partners. Perhaps this is one of the reasons for the differences in the functioning of the HMGB1 and HMGB2 proteins.

#### 2.2.1. Acetylation

The HMGB1 is characterized by the presence of two Nuclear Localization Sequences (NLS) in the regions of 27–43 and 178–184 a.a. and two CRM1-dependent non-canonical nuclear export signals (NES), due to which HMGB1 continuously shuttles between the nucleus and cytoplasm of most cell types. However, the equilibrium is shifted almost completely towards the nucleus [56]. Studies of monocyte culture demonstrated that the total acetylation of lysines in NLS regions of HMGB1 during the anti-inflammatory response led to the migration of HMGB1 to the cytosol and its further accumulation in cytoplasmic vesicles due to the loss of their ability to re-enter the nucleus. The properties of these cytoplasmic vesicles are similar to those of secretory lysosomes [57], which are Ca^2+^-regulated secretory organelles involved in inflammatory and immune responses, secreting their contents into the external environment in response to triggering signals [58]. Exocytosis of HMGB1-containing secretory lysosomes is triggered by lysophosphatidylcholine (LPC), a lipid produced by phospholipase A2 at the site of inflammation [57]. Thus, the acetylation of HMGB1 lysines in the NLS regions is directly related to the localization of the protein and its involvement in the signaling of the anti-inflammatory response. At the same time, it has been noted that the acetylation of HMGB1 lysines in the NLS regions is typical only in the case of active secretion of the protein and it is not typical for the protein in passive secretion [56].

According to the obtained data, HMGB1 of the calf thymus is characterized by the absence of acetylation in the NLS regions, which indicates a predominantly nuclear localization of the protein. Since HMGB2 has identical sequences of NLS regions and no acetylation sites were found in these regions, it can be assumed that HMGB2 is also predominantly localized in the nuclei.

Bonaldi et al. [56] also analyzed modifications of HMGB1 lysines, which are potentially susceptible to acetylation. It was shown that K50, K57, K59, K65, K68, K77, K82, K87, K88, K90, K141, K146, K147, K150, K152, K154, K163, and K165 in differentiated cells do not undergo acetylation, which agrees very well with our data. However, we have shown that calf thymus HMGB1 is also characterized by the presence of acetylation at positions K68, K82, and K152. HMGB1 acetylation at position K82 was also described by Pasheva et al. [59]. The authors showed that in HMGB1, K3 is acetylated at the N-terminal region, while the protein lacking the C-terminal region had an additional K82 acetylation site located in the linker. Perhaps the difference in K82, K68, and K152 acetylation is due to the structural characteristics of the protein. More recent studies have shown that HMGB1 can adopt at least two conformations [17]. In the first one, the negatively charged C-terminal sequence of HMGB1 is located between positively charged regions of two DNA-binding domains stabilizing the compact “closed” structure of the protein [60]. This conformation was attributed to the “inactive” state of the protein and characterized by the interaction between negatively charged C-terminal region (186–215 aa) with arginines 73 and 163, lysines 82 and 165, and isoleucine 159 [24]. This state is most likely characterized by the absence of acetylation at positions K68, K82, and K152, since these amino acid residues are located in close proximity to the site of interaction of the C-terminal region of the protein with the linker region and DNA-binding domains and their acetylation would prevent this interaction. The second conformation, on the other hand, is characterized by a disruption of the interaction between the C-terminal region of the protein with its DNA-binding domains, resulting in an “opened” conformation, attributed to the “active” state of the protein. This conformational transition might be induced, among the other factors, by the acetylation at positions K68, K82, and K152, which would also explain the conflict with the earlier results on the acetylation of these lysines [56]. The conformational variety of HMGB1 modulates its ability to interact with biological molecules and, thus, is very important from the functional point of view [17,60,61,62].

Our analysis of the HMGB2 mass spectra did not reveal acetylation at K50, K57, K59, K65, K68, K77, K82, K87, K88, K90, K141, K146, K147, K150, K154, K163, and K165. However, we identified an HMGB2 acetylation site at position K152. To our best knowledge, HMGB2 acetylation at these positions is reported for the first time.

Analysis of the earlier published data [56] indicates that in HMGB1, some lysine residues located in the N-terminal region (K3, K7, K8, and K12) and in the C-terminal region (K170, K172, K173, and K177) very rarely can be acetylated. We did not find lysine acetylation sites at positions K3, K7, K8, K12, K170, K172, K173, and K177 in either HMGB1 or HMGB2 under our experimental conditions. When HMGB1 binds to other biological molecules, the interaction can occur not only through DNA-binding domains but also with the participation of the N-terminal and C-terminal regions. PTMs in these regions, in particular, acetylation/deacetylation and methylation, are able to modulate protein–protein and DNA–protein interactions of HMGB1/2. Known examples of such modulation are the acetylation of HMGB1 at K3 [59] and acetylation of K12 [3], which are important in the interaction of HMGB1 with supercoiled DNA regions and branched DNA structures such as Holiday junction.

#### 2.2.2. Methylation

We have identified the following HMGB1 methylation sites: K7, R24, K59, K65, K87, K88, and K154. The presence of methylation at positions R24, R73, K76, K82, K85, K86, K141, K147, and K154 was also shown for HMGB2. Most of the identified methylated sites, namely K76, K141, K147, and K154, are highly conserved and found for both proteins HMGB1 and HMGB2. Their location, except for the two sites K7 and R24 described above, coincides with the position of non-acetylated lysines (see above [56]). Probably, methylation of lysines at these positions prevents their modifications by histone acetyltransferase (HAT) and is the primary reason for the lack of acetylation at these sites. The charge conservation of this region may be important for the interaction of the HMGB domain with DNA.

The rest of the HMGB2 methylation sites are located mainly near the linker site between the two DNA-binding domains. It is important to note that the last few amino acid residues of the A-domain of proteins and the linker site itself between the DNA-binding domains are slightly different for HMGB1 and HMGB2. Unlike HMGB1, in HMGB2, quite a lot of methylation sites are observed in this region (R73, K76, K82, K85, K86). Considering the importance of the structural organization of HMGB1 (and the presence of acetylation at position K82 in the HMGB1 linker region) for the functional activity of the protein, it can be assumed that the accumulation of methylation sites identified by us in the HMGB2 linker region may also have a functional significance.

According to the earlier published data, HMGB1 can be monomethylated at position K112 [63], which is attributed to the cytoplasmic localization of the protein. It has also been shown that methylation of K42 leads to conformational changes in the HMGB1 polypeptide chain [31]. It has been suggested that methylation at both K112 and K42 results in a decreased affinity of HMGB1 to DNA, which leads to the accumulation of the protein in the cytosol due to passive diffusion. The structure of the HMGB DNA-binding domains is characterized by the presence of three α-helices forming an L-shaped structure. This structural organization is stabilized by strong hydrophobic interactions at the corner apex [11]. K112 is highly conserved in HMGB1 and HMGB2 of vertebrates. Methylation at K112 affects the formation of this hydrophobic core by changing the orientation of the α-helix I.

According to our data, calf thymus HMGB1 and HMGB2 are characterized by the absence of methylation both at position K42 and at position K112. The absence of methylation at these sites can be considered an indirect confirmation of the predominantly nuclear localization of the studied proteins.

#### 2.2.3. Phosphorylation

Along with acetylation and methylation, it was reported that phosphorylation of serine residues located in close proximity to the NLS1 and NLS2 regions of HMGB1 (S34, S38, S41, S45, S52, and S180) blocked its re-entry into the cell nucleus [64]. It was also shown that the accumulation in the cytoplasm of RAW264.7 cells and human monocytes is followed by the secretion of the protein into the intercellular space. Since HMGB1 lacks a secretory signal peptide and does not pass through the endoplasmic reticulum and the Golgi complex, it is likely that the secretion is accompanied by the formation of the previously described cytoplasmic vesicles [65].

Based on the analysis of the mass spectra, we have identified the following HMGB1 phosphorylation sites: T22, T85, and Y155. The functional consequences of the HMGB1 phosphorylation at these sites are still unknown. One can assume that T22, T85, and Y155 phosphorylation should not affect the protein localization in the cell due to the remoteness of these sites from the NLS1 and NLS2 of HMGB1. However, these PTMs may affect both the DNA-binding properties and the interaction of HMGB1 with other proteins.

We have identified a number of phosphorylation sites within the HMGB2 protein at positions Y71, Y78, Y144, Y155, and Y162. The biological role of these modifications also has not been established yet. Their location affects the region near the linker site between the domains and the B domain. Based on the arguments similar to those for HMGB1, one can assume that phosphorylation of these amino acid residues should not lead to changes in the cellular localization of HMGB2, but it might affect the nature of the interaction of HMGB2 with DNA and other proteins.

#### 2.2.4. Glycosylation

It has previously been shown [4,66] that several asparagine residues of HMGB1, namely N37 or residues N134 and N135, that are located in the Asn-Xxx-Ser/Thr N-glycosylation motif (where Xxx stands for any amino acid, except for proline), can be N-glycosylated. It has also been shown that HMGB1 glycosylation is necessary for the interaction of the protein with CRM1, also known as exportin 1, XPO1, and chromosomal region maintenance 1 protein. CRM1 mediates the nuclear export and further secretion into the extracellular space of various proteins and RNAs. When analyzing HMGB1 and HMGB2 isolated from the calf thymus, we have not identified any glycosylation sites, which is also an indirect confirmation of the predominantly nuclear localization of the studied proteins.

#### 2.2.5. ADP-Ribosylation

Similar to the HMGB1 modifications described above, ADP-ribosylation also affects protein localization within the cell. Mono- and poly-ADP-ribosylation is required for nuclear export and release of HMGB1 upon cell death, especially during necrosis [67,68]. No ADP-ribosylation sites were found in the HMGB1 and HMGB2 samples studied in this work.

#### 2.2.6. Oxidation

One of the key modifications of HMGB1 that directly affects the functional properties of the protein is the oxidation of cysteines at positions C23, C45, and C106. Translocation of the HMGB1 protein from the nucleus to the cytoplasm is accompanied by the oxidation of C23 and C45, which form a disulfide bond, while C106 remains reduced [29,69]. In addition, the oxidation of these residues is responsible for the binding of HMGB1 to a particular receptor (receptor for advanced glycation endproducts, tall-like, etc.), and during secretion, the oxidative status of the protein determines its activity associated with autophagy and immunity [69,70]. It should be noted that all three HMGB1 cysteines are completely reduced in the cell nucleus. In a recent study, it was shown that when C23 in the nucleus are oxidized under the action of reactive oxygen species, which primarily affects protein binding to DNA [71,72], protecting it from hydrolysis under oxidative stress.

According to our data, HMGB1 and HMGB2 of the calf thymus are characterized by oxidized cysteine at the C23 position. C45 and C106 are present in a reduced form, which does not allow the formation of a disulfide bridge between these cysteine residues and probably does not affect the localization of the protein in the cell, but may affect the ability of proteins to bend DNA double helix [18,71] and interact with non-canonical DNA structures [3,4,9,13].

Post-translational modifications of HMGB1 and HMGB2 proteins affect not only their functioning in chromatin but also determine the localization of proteins in the cell. Our analysis of the mass spectra showed that, despite the high similarity of the amino acid sequence, the nature of the PTM distribution in HMGB1 and HMGB2 is different. When studying the modifications of HMGB1 and HMGB2, we did not reveal acetylation sites at protein positions K3, K7, K8, K12, K170, K172, K173, and K177, which, in general, taking into account the frequency of modification in this region, is consistent with those described for HMGB1 data in the literature. For HMGB2, we also showed the presence of potential methylation sites at positions R24, R73, K76, K82, K85, K86, K141, K147, and K154. Most of the identified methylated sites, namely K76, K141, K147, and K154, are highly conserved in HMGB1 and HMGB2. Their arrangement coincides with that of the lysines described in the literature as «can’t be acetylated». The remaining potential HMGB2 methylation sites that we have identified are located mainly near the linker site between two DNA-binding domains. It is important to note that the region of the A domain of both proteins adjacent to the linker sequence and the region itself between the HmgB1 and HMGB2 domains are not highly conserved. Unlike HMGB1, in HMGB2, quite a lot of methylation sites are observed in this region (R73, K76, K82, K85, K86). In addition, we identified a number of HMGB2 phosphorylation sites at positions Y71, Y78, Y144, Y155, and Y162. As in the case of HMGB1, the biological role of these modifications has not yet been revealed. The location of pY71, pY78, pY144, pY155, and pY162 affects the region near the linker site between the domains and the B domain. Due to the remoteness of the modified region from the NLS1 and NLS2 regions, as in the case of HMGB1, we can assume that phosphorylation of these amino acid residues should not lead to a change in the protein localization in the cell but may affect the nature of the interaction of HMGB2 with DNA and other proteins. Considering the importance of the spatial configuration of the HMGB1 polypeptide chain (and the presence of acetylation at the K82 position on the HMGB1 linker site) for the functional activity of the protein, it can be assumed that the high number of methylation sites that we identified in the region of the HMGB2 linker site may also have a functional significance.

Potential modification sites for the HMGB1 protein, which we have identified for the first time, are located mainly in its DNA-binding A-domain and in the linker region between two HMGB domains. On the contrary, HMGB2 PTMs are concentrated in the B-domain of the protein and also in the linker region, but the modifications found within the linker regions of the two proteins are different. Taking into account all the results obtained, it can be assumed that, despite the high homology of the amino acid sequence and spatial organization of the HMGB1 and HMGB2 proteins, the manner of their binding to DNA and other macromolecules can be different.

### 2.3. Interaction of HMGB1 and HMGB2 Proteins with DNA

To compare mechanisms of interaction of DNA with HMGB1 and HMGB2 proteins, we applied circular dichroism spectroscopy. The optical activity of the HMGB1/DNA and HMGB2/DNA complexes was measured in 15 mM NaCl solution depending on protein/DNA ratio r (*w*/*w*) in the system (Figure 4). The 15 mM NaCl corresponds to the ionic strength in the cell nucleus, where the HMGB-DNA complexes are formed. The spectra of the complexes clearly demonstrate two regions on either side of the 235 nm mark. To the left of this point (in the region of shorter wavelengths), the spectra represent a superposition of the optical activity of the protein and the DNA in the complex. In the region of wavelengths above 235 nm, CD reflects only the optical activity of DNA. Increasing protein content in the system leads to an increased CD signal, forming deeper minima in the range of 200–230 nm and increasing CD in the range of 260–300 nm. The latter resembles PSI (Polymer and Salt Induced) type of CD spectra [73,74], typical for large supramolecular complexes of DNA with nuclear proteins [75,76,77,78,79]. Despite some similarity in the spectra, their analysis reveals a noticeable difference in the spectral characteristics of the complexes with HMGB1 and HMGB2.

As shown earlier, the interaction of HMGB1 with DNA is characterized by the formation of large supramolecular complexes at sufficiently high protein/DNA ratios in the system [21,22,52]. Such complexes are usually formed due to the simultaneous binding of two different protein regions with two different targets, forming a kind of intermolecular crosslinking. In contrast to the HMGB1 protein, the binding of HMGB2 to DNA leads to the appearance of the s PSI type CD spectrum at a relatively low protein content in the complex r = 0.75. It should be noted that although the change in the shape of the CD spectra reflects an increase in light scattering in solution rather than the structural properties of DNA, the PSI-type CD spectra remain a good indicator of supramolecular complexes formed in solution. Similar spectral changes occur in the HMGB1/DNA complexes at significantly higher protein-to-DNA ratios (r > 1). Thus, one may conclude that despite the high homology of the proteins, HMGB2 easily forms large multimolecular complexes. As mentioned above, the main difference in the primary structures of the HMGB1 and HMGB2 proteins is the length of the C-terminal acidic fragment. The shorter HMGB2 region is unable to form a stable “closed” conformation, allowing the negatively charged region and both HMGB domains to participate in the formation of stable multimolecular complexes. In the HMGB1 protein, the negatively charged tail is long enough to remain bound to one of the positively charged DNA-binding domains, thereby excluding that HMGB domain from interaction with DNA and the acid tail from interactions with other proteins.

## 3. Materials and Methods

### 3.1. Proteins and DNA

Nuclear proteins HMGB1 (M = 26,500 Da) and HMGB2 (M = 26,000 Da) were isolated from calf thymus by extraction with 5% perchloric acid followed by precipitation with 6 volumes of acidified acetone at −20 °C, as described previously [21]. Further purification of protein fractions was carried out using ion-exchange chromatography in an FPLC system (Pharmacia BioTech, Uppsala, Sweden) using a Mono Q column. Proteins were eluted in a linear gradient in 0.05 M Tris-HCl buffer pH 7.4 within 0.03–0.83 M NaCl. The HMGB2 protein is released at 0.43 M NaCl, and the HMGB1 protein is released at 0.5 M NaCl. After the purification of the proteins, the electrophoretic analysis of the fractions was performed. Protein fractions were pooled according to the results of the analysis. The proteins were then precipitated with acetone and dried under vacuum, as described above. Ready protein preparations were stored at −20 °C. Electrophoretic analysis of proteins was carried out with Laemmli SDS PAGE [80]. The electropherogram is shown in Appendix A. Protein purity was checked with SDS PAGE [80] (Appendix A). Protein concentration was determined spectroscopically, as described elsewhere [16].

High molecular weight calf thymus DNA (Sigma, St. Louis, MO, USA) was used for spectroscopic experiments. The DNA concentration and nativeness were determined spectrophotometrically [81].

### 3.2. Enzymatic Hydrolysis and MALDI-FT-ICR-MS Analysis

Samples for the identification of PTMs were prepared according to the approach described earlier [82,83]. Proteins extracted from the thymus were separated with Laemmli SDS PAGE [80] (Appendix A). The isolated proteins were identified with Western blot using antibodies ab 79823, ab 67282 (Abcam, Cambridge, UK) against HMGB1 and HMGB2, respectively. The strips cut from the polyacrylamide gel corresponding to the HMGB1 and HMGB2 proteins were treated with a solution of 40% acetonitrile in 0.1 M ammonium bicarbonate at 37 °C for 15 min, followed by enzymatic trypsin hydrolysis (4 h, 37 °C) directly in the gel. The trypsinolysis reaction was stopped by adding 0.5% and 10% of trifluoroacetic acid and acetonitrile, respectively. Samples for MALDI mass spectrometry were obtained by co-crystallizing a solution of the trypsinolysis products with a solution containing 20 mg/mL of 2,5-dihydroxybenzoic acid in 30% acetonitrile, 0.1% trifluoroacetic acid, followed by air-drying.

### 3.3. Mass Spectrometry

Analysis of the PTMs of the HMGB1 and HMGB2 proteins was carried out using MALDI-FT-ICR-MS according to the peptide mass fingerprinting scheme. To do this, after the electrophoresis, the gel fragments containing HMGB1 or HMGB2 proteins were minced and treated with 40% acetonitrile in 0.1 M ammonium bicarbonate at 37 °C for 15 min. Trypsin hydrolysis in the gel was conducted for 4 h at 37 °C. To stop the reaction, 0.5% TFA and 10% acetonitrile were added to the reaction mixture. The sample for the MALDI-FT-ICR-MS (matrix-activated laser desorption/ionization Fourier transform ion cyclotron resonance mass spectrometry) analysis was obtained by mixing 600 nL of solution of protein fragments and 300 nL of 2,5-dihydroxybenzoic acid (20 mg mL^−1^ in 30% aqueous acetonitrile, 0.1% TFA) and dried in air. Mass spectra were registered in the positive ion mode within the range of 500–3000 m/z using the MALDI-FT-ICR mass spectrometer (Varian 902-MS). The mass spectra of the proteins were analyzed using Mascot software (Matrix Science, London, UK, www.matrixscience.com, accessed on 20 February 2019). Additionally, Protein Prospector MS-Fit software was used to identify the proteins and their post-translational modifications [84,85]. Mass spectra were searched against protein sequences from the UniProt knowledge database (www.uniprot.org, accessed on 20 February 2019) using the mammals search engine. Spectra were searched with: monoisotopic mass values; 1 + peptide charge state; mass tolerance of 10 ppm; strict trypsin specificity; and allowing up to four missed clearance sites. The task of these programs is to correlate the obtained mass spectra with protein sequences according to a certain algorithm and evaluate the degree of agreement between the experimental and calculated spectra. This estimate is always probabilistic and, therefore, never reaches 100%. Appendix A presents experimental and theoretical values of the ratio of the mass of the peptide to its charge m/z, the intensity of the band in spectrum I, the identified potential post-translational modifications, the initial and final position of this peptide in the amino acid sequence of the protein, and, finally, the amino acid sequence of this peptide. The reliability of the interpretation of the spectrum was evaluated using the value of Δm/z, which shows the deviation of the real spectrum from the theoretical one. The smaller this deviation, the more likely the correct correlation of peaks in the mass spectrum. In all determined lines in the spectrum, the deviation of the measured masses from the theoretical values does not exceed 0.0030 a.m.u. Further analysis included manual data processing, analysis of protein amino acid sequence coverage/overlapping by peptides, correlation of potential modification sites identified by us with previously published data from other authors, and construction of a map of protein PTM location. Biological samples were analyzed in 2 biological and 2–3 analytical replicates.

### 3.4. Western Blotting

Cellular extracts containing HMGB1 and HMGB2 proteins were resuspended in an equal volume of 2X lysis buffer (50 mM Tris-HCl (pH 6.8) containing 10% glycerol and 1% SDS) and heated to 98 °C for 10 min. The protein mixture (5–30 μg of total protein per well) was separated with Laemmli SDS PAGE [80], and obtained proteins were transferred to an Amersham Hybond-ECL nitrocellulose membrane (Thermo Fisher Scientific, Waltham, MA, USA) according to the standard protocol of Western blotting technology. After that, the membrane was blocked with a solution of 5% skimmed milk powder, 0.1% Tween-20 in PBS for 60 min at room temperature. Hybridization of the membrane in solution with I-antibody in 1% milk, 0.1% Tween-20 in PBS was carried out overnight at 4 °C. After that, the membrane was washed 3 times for 5 min with 0.1% Tween-20 in PBS and incubated with solutions of the corresponding secondary antibodies conjugated with peroxidase and diluted in 1% milk, 0.1% Tween-20 in PBS for 1 h at room temperature. We used Anti-HMGB1 antibody ab 79823 (Abcam, Cambridge, UK), Anti-HMGB2 antibody—Chip Grade ab 67282, and Peroxidase AffiniPure F(ab’) Fragment Goat Anti-Rabbit IgG (H + L) no. 111-036-045 (Jackson Immunoresearch, West Grove, PA, USA). After washing three times with PBS, the membrane was analyzed using SuperSignal West Dura Extended Duration Substrate (Thermo Fisher Scientific, Waltham, MA, USA) and Chemidoc Touch Imaging System (Bio-Rad, Hercules, CA, USA).

### 3.5. Circular Dichroism Spectroscopy

DNA complexes with HMGB1 and HMGB2 proteins were prepared in a 15 mM NaCl solution by mixing equal volumes of DNA and protein solutions of appropriate concentrations to achieve protein-to-DNA ratios (*w*/*w*) r in the range of (0 ≤ r ≤ 1.5). Final concentration of DNA in the complex is 20 mkg/mL. Protein circular dichroism (CD) spectra were obtained using a Cary-60CD polarimeter-dichrograph (Beckman, United States) in cylindrical quartz cells with an optical path length of 0.5 cm in the range of 200–250 nm. The CD spectra of DNA–protein complexes were obtained using a Mark V dichrograph (Jobin Yvon, Longjumeau, France) in the range of 200–320 nm. The devices were calibrated with a solution of D-camphorosulfonic acid in the region of 190 and 290 nm. The measurement accuracy was ±0.0004°. The isotropic absorption of all CD samples was monitored.

The obtained CD spectra are presented as the difference between the absorptions of left- and right-polarized light ∆A (∆A = A_L_ − A_R_) for the complexes and in terms of molar ellipticity [*θ*] for protein solutions. The values of the molar ellipticity of proteins were calculated based on the average weight of the amino acid residue, which is equal to 102.31 and 100.39 for HMGB1 and HMGB2, respectively.

In cases where it is necessary to analyze the spectra of a multicomponent mixture, the use of specific values (for example, molar ellipticity [*θ*]) is not quite convenient. This is due to the fact that there may be regions in the spectra where the obtained signal is a superposition of contributions from individual components in the mixture. In our case, there is an overlap of the spectral bands of protein and DNA at wavelengths less than 250 nm. The degree of α-helicity of the proteins was estimated based on two independent approaches. First, using the K2D3 algorithm [53,54]. Second, using the equation suggested by Morrow et al. [55] based on the value of molar ellipticity at 222 nm [*θ*_222_]:*α*(%) = (−[*θ*_222_] + 3000)/39,000(1)

## 4. Conclusions

The functions of proteins are closely related to their conformational state. To study the conformational features of the HMGB1 and HMGB2 proteins and the structure of DNA–protein complexes, we used circular dichroism (CD) and UV absorption spectroscopy. This approach makes it possible to detect differences in the secondary structure of protein molecules and changes in the structural organization of DNA/protein complexes. As mentioned above, HMGB-domain proteins are often compared with natively unfolded proteins [3,9,19], and they are able to change their conformation upon binding to other biological molecules. Such structural flexibility of the polypeptide chain plays an important role in the formation of multicomponent functional complexes in chromatin [47,48]. The studies of the conformational transitions in HMGB-domain proteins demonstrated that there is a dynamic equilibrium between active “opened” and inactive “closed” forms of the protein [24,25,60]. This conformational transition can be associated with PTMs of the protein and/or by its interactions with other molecules [25,60,79].

The PTMs of HMGB1, which we have identified, are located mainly in the DNA-binding A-domain and in the linker region between two HMGB domains of the protein. On the contrary, the majority of HMGB2 PTMs are found within the B-domain of the protein. There are also some modifications in the linker region of HMGB2; however, they are different compared to HMGB1. It can be expected that the number of modifications is higher in the functionally significant regions of the protein. Hence, it can be assumed that a greater number and variety of modifications within the B-domain of HMGB2 reflects its greater functional activity as compared to the B-domain of HMGB1. HMGB2 has greater conformational flexibility; it demonstrates more effective DNA compaction and condensation compared to HMGB1; it has a greater number of PTMs identified in the B-domain. Altogether, the above results indicate that the B-domain of HMGB2 is involved in protein functioning to a much greater extent than the B-domain of HMGB1.

Both proteins are involved in various processes: replication, repair, etc. However, the presence in the cell of both proteins so close in their primary and secondary structures cannot be accidental. Combining the results of CD spectroscopy and mass spectrometry, one can conclude that, despite the high homology of the primary structure of the HMGB1 and HMGB2 proteins, the mechanisms of their binding to DNA and other proteins can differ significantly. Interacting with DNA, both HMGB1 and HMGB2 significantly bend the double helix. However, according to the earlier published data, a single DNA-binding domain of HMGB2 induces a bend of about 100°, while two HMGB1 DNA-binding domains together induce bends of less than 80° [86]. This might indicate that HMGB2 demonstrates stronger binding to DNA than HMGB1. Stronger DNA binding suggests a higher probability of nuclear localization. There is a huge pool of experimental data on the extracellular functions of HMGB1 [4,9], and when HMGB1 leaves the nucleus, one can expect that HMGB2 still remains bound to the chromatin.

Chromatin is a complex, highly dynamic system that includes a huge number of different factors. The role of proteins in a living cell is a set of manifestations of their various contacts with many partners. The DNA–protein systems studied in this work are an in vitro model system, which makes it possible to study the interaction of a particular protein with DNA using spectral methods in the absence of other partners. The differences that we have identified may indicate a difference in the functioning of these proteins in chromatin, which manifests itself in the difference in the spectral characteristics of not only the HMGB1 and HMGB2 proteins themselves but also their complexes with DNA. These features of proteins can affect the functioning of the chromatin-remodeling complex since it is the DNA loop formed during the interaction of DNA with HMGB-domain proteins that initiate the recruitment of this complex. This, in turn, can affect basic cellular processes, such as transcription.

We believe that it is important to understand the subtle structural differences between the HMGB1 and HMGB2 proteins, which undoubtedly affect the mechanisms of their interaction with DNA.

## Figures and Tables

**Figure 1 ijms-24-03577-f001:**
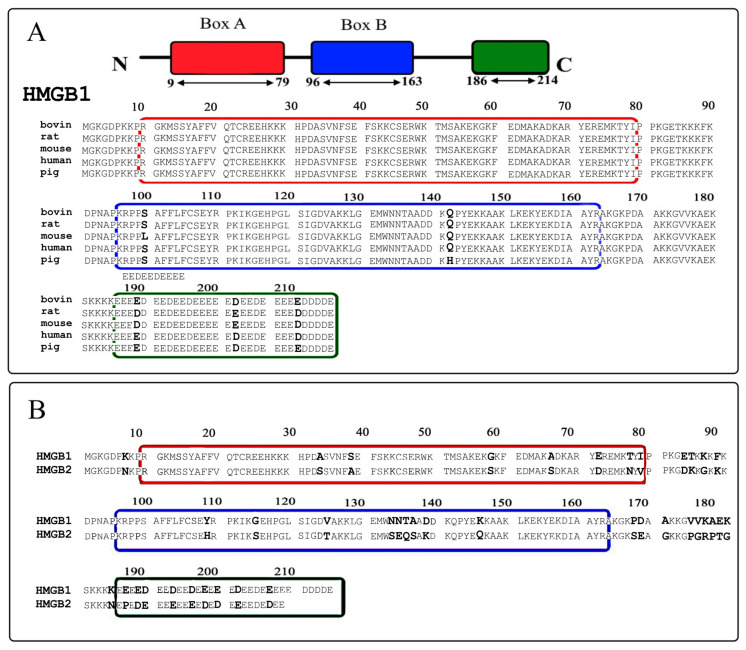
Primary structures of HMGB1 and HMGB2 proteins. **Panel A**: primary structures of the bovine, rat, mouse, human, and pig HMGB1 proteins. **Panel B**: primary structures of bovine HMGB1 and HGB2. The DNA-binding A- and B-domains are marked with red and blue boxes, respectively. Green box corresponds to the C-terminal acidic region. Amino acid substitutions, identified for different species, are marked with bold letters.

**Figure 2 ijms-24-03577-f002:**
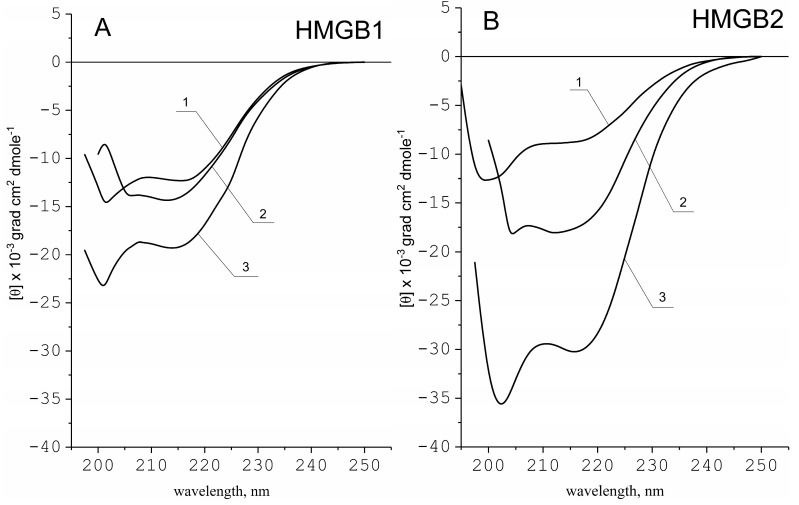
CD spectra of HMGB1 (**panel A**) and HMGB2 (**panel B**) proteins. Circular dichroism (CD) spectra of HMGB1 and HMGB2 are presented the Figure. The obtained CD spectra are presented in terms of molar ellipticity [θ]. The spectra are obtained in an aqueous solution (curve 1), 80% ethanol (curve 2), and 1.5 M NaCl (curve 3). The spectra are obtained in an aqueous solution (pH 6.0), 80% ethanol, and 1.5 M NaCl. Estimation of α-helicity (see Discussion) showing that in neutral solvent, HMGB1 forms ~25% of α-helices, while HMGB2 only ~15%. HMGB2 is characterized by a greater ability to form α-helices in 80% alcohol and 1.5 M NaCl. The degree of α-helicity of the proteins in 80% alcohol solution is 30% and 40% for HMGB1 and HMGB2, respectively; in solution of 1.5 M NaCl the α-helicity is ~40% and ~70%, respectively.

**Figure 3 ijms-24-03577-f003:**
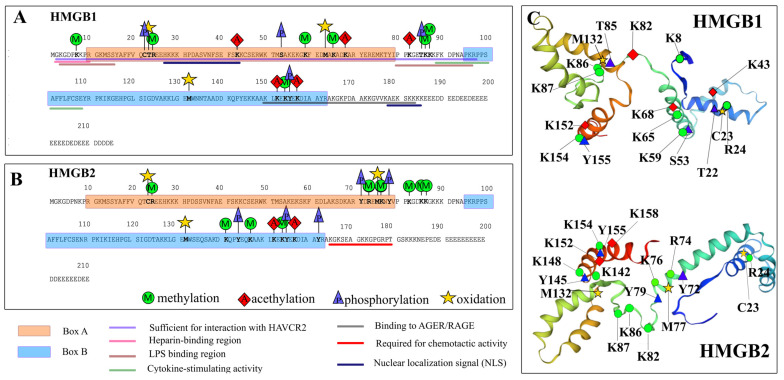
Post-translational modifications of calf thymus HMGB1 (**panel A**) and HMGB2 (**panel B**) non-histone proteins. The schematic representation of potential PTM site location (**panel C**) (the PTMs are also listed in the Table). Potential HMGB1 PTM sites that we have identified for the first time are located mainly in its DNA-binding A-domain and linker region connecting the A and B domains. On the contrary, HMGB2 PTMs are concentrated in the B domain and also within the linker region. However, the nature of the modifications within this region is different for HMGB1 and HMGB2, which might affect the functioning of these proteins.

**Figure 4 ijms-24-03577-f004:**
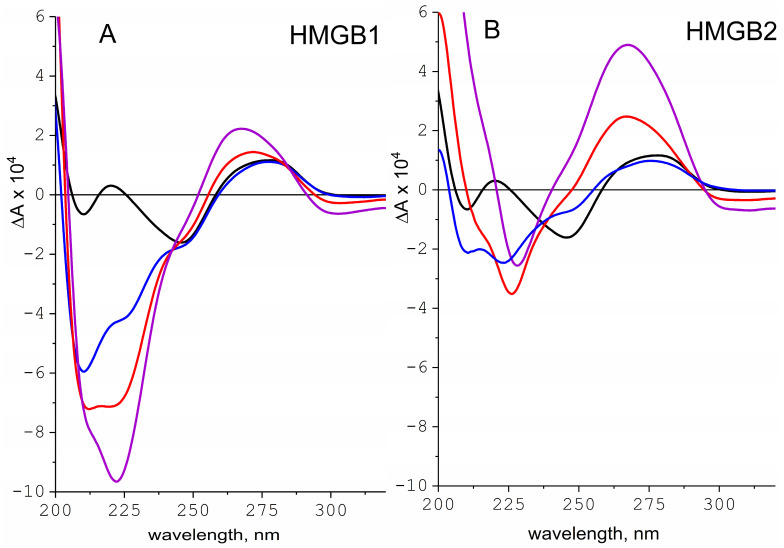
CD spectra of complexes HMGB1/DNA (**A**) and HMGB2/DNA (**B**) at different protein/DNA ratios in 15 mM NaCl. CD spectra of protein/DNA complexes at protein/DNA weight ratios r 0.5 (blue), 0.75 (red), and 1.0 (violet); DNA (black) in a 15 mM NaCl solution. The obtained CD spectra are presented as the difference between the absorptions of left- and right-polarized light ∆A (∆A = A_L_ − A_R_). At the same protein/DNA ratios, the CD spectra of the complexes are remarkably different. The shape of the CD spectra in the HMGB2/DNA complex indicates the presence of light scattering already at r = 0.75, which is typical for the formation of large supramolecular particles in solution, while for the HMGB1/DNA complexes, similar changes in the spectrum occur at r > 1.

**Table 1 ijms-24-03577-t001:** α-helicity (%) for HMGB1 and HMGB2 proteins in different solutions.

Protein	α-Helicity (%), [θ_222_]/K2D3 *		
	Water	80% Alcohol	1.5 M NaCl
HMGB1	25/30	30/30	42/46
HMGB2	15/14	40/42	70/70

* The α-helicity was estimated using equation from [55] and using the K2D3 algorithm [53,54].

**Table 2 ijms-24-03577-t002:** The results of MALDI mass spectrometry analysis of HMGB1 and HMGB2 from calf thymus.

Protein	Modification	Peptide	Sequence
HMGB1	1Oxidation	58–65	(K)GKFEDMAK(A)
	1Methyl 1Oxidation	58–65	(K)GKFEDMAK(A)
	2Acetyl 1Methyl 1Phospho	151–157	(K)LKEKYEK(D)
	1Acetyl 1Methyl 1Oxidation	60–70	(K)FEDMAKADKAR(Y)
	1Acetyl 1Phospho	77–87	(K)TYIPPKGETKK(K)
	2Phospho	77–87	(K)TYIPPKGETKK(K)
	1Oxidation	13–24	(K)MSSYAFFVQTCR(E)
	1Acetyl	31–48	(K)HPDASVNFSEFSKKCSER(W)
	1Oxidation	129–146	(K)LGEMWNNTAADDKQPYEK(K)
	2Methyl 1Oxidation 1Phospho	8–24	(K)KPRGKMSSYAFFVQTCR(E)
	1Methyl 1Phospho	13–29	(K)MSSYAFFVQTCREEHKK(K)
	1Oxidation	58–65	(K)GKFEDMAK(A)
	1Oxidation	71–76	(R)YDREMK(N)
HMGB2	1Phospho	164–172	(R)AKGKSEAGK(K)
	2Acetyl 1Methyl 1Phospho	151–157	(K)LKEKYEK(D)
	1Dimethyl 2Methyl	77–86	(K)NYVPPKGDKK(G)
	1Oxidation	129–139	(K)LGEMWSEQSAK(D)
	1Acetyl 2Dimethyl 1Phospho	77–86	(K)NYVPPKGDKK(G)
	1Oxidation	128–139	(K)KLGEMWSEQSAK(D)
	1Oxidation	13–24	(K)MSSYAFFVQTCR(E)
	1Acetyl 2Methyl 1Phospho	140–150	(K)DKQPYEQKAAK(L)
	1Dimethyl 1Methyl 2Phospho	71–82	(R)YDREMKNYVPPK(G)
	1Methyl 1Phospho	13–29	(K)MSSYAFFVQTCREEHKK(K)

## Data Availability

The data presented in this study are available in the article and Appendix A.

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
