# Peer review of "Structural Characteristics of High-Mobility Group Proteins HMGB1 and HMGB2 and Their Interaction with DNA"

_ijms, 2023, doi:10.3390/ijms24043577_

Round 1
Reviewer 1 Report
The aim of this work is to characterize the structural organization and DNA-binding capacity of HMGB1 and HMGB2 proteins, which are non-histone proteins relevant to many biological processes. To this end, the authors used UV circular dichroism (CD) spectroscopy and MALDI mass spectrometry. They conclude that, despite the similar primary structures between HMGB1 and HMGB2, they have quite different post-translational modifications (PTMs) and secondary structures. The results are succinct, with no functional validations or differential prediction of binding, but clear and interesting, and useful for future research on HMGB proteins. IJMS articles usually have more data and deeper characterization of mechanisms rather than pure characterizations of molecules. Therefore, this article's publication in IJMS may rely on an editorial decision.
I want to point out some aspects of the text that can be improved.
1) ABSTRACT:
· “Non-histone nuclear proteins HMGB1 and HMGB2 are involved in many biological pro-10cesses, such as replication, transcription, repair, etc.”
Comment: eliminate “etc”.
· “ their complexes with DNA was.....”
Comment: It should be changed by “their complexes with DNA were....”
· “The HMGB1 PTMs are located predominantly in the DNA-binding A-domain and linker region connecting the A and B domains.”
Comment: The authors should briefly explain what are the A and B domains in the abstract.
2) INTRODUCTION:
· “Such features can comprise”
Comment: Take the extra space between “can” and “comprise”
· “… where it stimulates the immune response and functions as a signaling molecule in response to damage of cell integrity, necrosis, etc….”
Comment: eliminate “etc”.
· “Depending on the redox state, HMGB1 can (1) act as a signaling molecule 67 by activating the MAPKs, NF-kB and phosphoinositide-3-kinase/AKT signaling path-68ways, (2) take part in the regulation of cell migration, or (3) participate in the immune 69response and synthesis of anti-inflammatory cytokines [4,18,22,24,28-30].
Comment: Here, instead of just citing the papers [4,18,22,24,28-30], the authors should say in what contexts this happens (e.g. cancer cells?, heart disease? Normal physiological conditions?)
3) FIGURES
· Design of figure 1 can be improved, namely by changing the Font of the alignment to “Courier”
· Design of figure 3 can be improved (e.g. the lettering of the amino acid sequence is hard to see, as well as the captions in the third panel, panels should be labeled as A, B, and C and the legend should be modified according to that.
4) METHODS
· Some methods sections are in italic.
Reviewer 2 Report
In this paper the authors report a characterization of HMGB1 and HMGB2 proteins isolated from calf thymus. The paper is interesting but there are a number of points which must be addressed:
1) the alignment of HMGB1 and HMGB2 rather than conservation among HMGB1 from different sources should be shown. This would help to clearly identify differences in the sequence between the two proteins.
2) methods must be described in much more detail, in particular protein purification and preparation of samples for CD. Indicate pH of the 1.5 M and 15 mM NaCl solutions employed, why not use buffered solutions?
3) the first paragraph of the Results section should be expanded indicating the source of the protein and briefly the purification procedure instead of citing Methods and SI sections.
4) how quantitative analysis of CD spectra is performed is unclear. I suggest using the Dichroweb site for analysis of the spectra of the proteins. Is the difference in alfa-helical content significant?
5) please justify why 15 mM NaCl has been used for the study of HMGB and DNA interaction, this low salt concentration could lead to artifactual binding.
4) table 1 is quite difficult to read, where are modifications found in the various peptides?
5) minor points: in table 1 part of the title is missing, there are a few typos (e.g. line 149 conservation not conservatism). Parts of the methods section are in italics.
Reviewer 3 Report
The manuscript submitted by Starkova et al. characterizes the post-translational modifications (PTMs) of calf-thymus high-mobility group B proteins (HMGBs) and their secondary structure in the native state and the presence of α-helix inducers and naked DNA. The study has several major issues that the authors should address before it could be considered for publication in IJMS. Below, you can find a detailed explanation:
- The title should contain high-mobility group proteins instead of non-histone chromosomal proteins.
- The introduction must contain references to the three main types of high-mobility group proteins: HMGA, HMGB, and HMG and summarize their similarities and differences.
- Purity of HMGB1 and HMGB2. The methods section should include the experimental procedure to separate both HMG variants. Supplementary figure S1C is a western blot analysis, which does not support the purity of each variant, necessary condition for the CD studies.
- The equation used to calculate the amount of α-helix from the CD spectra should appear in the materials and methods section.
- A table with the alpha helix values calculated for each condition must appear in Figure 2. Showing the secondary structure prediction for each protein could support the results.
- The expression blue-shift is incorrect, as it refers to a shift of the minimum/ maximum to lower wavelengths, not to the amount of CD signal.
- The more important concern of this study is regarding the study of the HMGBs PTMs because the assignment of the PTMs to specific residues is unjustified, as the analysis did not include tandem mass spectrometry in the experimental design. In most of the peptides shown in table 1, the identified PTMs could be in several residues. For acetylation and methylation, peptides have more than one lysine residue. For oxidation, in addition to cysteine, peptides contain methionine, another amino acid prone to oxidation. Finally, for phosphorylation, most peptides also have more than one S, T, or Y, which could be phosphorylated. Therefore, the unambiguous assignment of the PTMs cannot be accomplished.
- A more detailed explanation of the analysis of the PTMs by mass spectrometry must appear in the methods section.
- In the results and discussion sections, the relevant findings of this study should be explained clearly, separated from the results from other authors.
- The possible changes in the interaction between HMGBs and naked DNA and those established when bound to chromatin, should be discussed.
Round 2
Reviewer 2 Report
The authors have adequately addressed my comments. Please add a reference regarding ionic strength in the nucleus (line 421)
Reviewer 3 Report
The revised version of the manuscript has improved considerably, addressing all the concerns raised during the initial review. Therefore, the manuscript is now suitable for publication in IJMS.